



# The six rights of how and when to test for soil C saturation

Johan Six[1], Sebastian Doetterl[1], Moritz Laub[1], Claude R. Müller[1], Marijn Van de Broek[1]

[1]Department of Environmental Systems Science, ETH Zurich, Zurich, 8092, Switzerland

*Correspondence to*: Johan Six (jsix@ethz.ch)

**Abstract.** The concept of soil organic carbon (SOC) saturation emerged a bit more than 2 decades ago as our mechanistic understanding of SOC stabilization increased. Recently, the further testing of the concept across a wide range of soil types and environments has led some people to challenge the fundamentals of soil C saturation. Here, we argue that to test the concept, one should pay attention to 6 fundamental principles or rights (R's): the right measures, the right units, the right fractionation

method, the right soil type, the right mineralogy, the right saturation level. Once we take care of those 6 rights across studies, we find a maximum of C stabilized by minerals and estimate based on current data available that this maximum stabilization is around $82 \pm 4$ g C kg$^{-1}$ silt+clay for 2:1 clay dominated soils while most likely only around $46 \pm 4$ g C kg$^{-1}$ silt+clay for 1:1 clay dominated soils. These estimates can be further improved using more data, especially for different mineralogies across varying environmental conditions. However, the bigger challenge is on how and which C sequestration strategies to implement

in order to effectively reach this 82/46 g C kg$^{-1}$ silt+clay in soils across the globe.

In recent years, several studies (e.g., Begill et al. 2023; Salonen et al. 2023) have questioned the concept of soil carbon (C) saturation, i.e., organic C stabilized by soil minerals (Hassink, 1997; Six et al. 2002). Here, we want to draw attention to six fundamentals that we think one should be cognizant about when "testing" and "questioning" soil C saturation.

## 1 The right measures

Soil C saturation is by definition a non-responsiveness of soil C content upon an increase of C inputs and thus can be assessed by plotting soil C (or any fraction of it, e.g., mineral-associated organic C (MAOC)) versus C input (Six et al. 2002). However, because of (1) confounding factors (e.g., texture, mineralogy, climate, etc.) when going across systems/biomes, (2) the difficulties to determine accurately C inputs (especially to specific fractions of soil C), and (3) the limited range in C inputs

within one system/experiment, plots of soil C versus C input are often not sensible, nor feasible and thus inconclusive (see West & Six 2007; Stewart et al. 2007; Feng et al. 2014). Specifically for fractions of soil C, in Stewart et al. (2007) it was correctly argued that saturation can also be tested by plotting a fraction of soil C, such as MAOC, versus total SOC. However, this does not take care of the above-mentioned confounding factors when going across systems/biomes, nor increase the range of "C inputs" within one system/experiment. Thus, the most elegant, accurate and thus preferable way to test for saturation is

by plotting MAOC versus silt+clay content in the right units.



## 2 The right units

As for any analysis, using the correct units is of great importance. For C saturation, the best data analysis that can be done is relating MAOC (y-axis) expressed on a per unit total soil (i.e. g C kg$^{-1}$ total soil or mg C g$^{-1}$ total soil; the emphasis on per *total soil, which is the < 2 mm fine soil)* versus silt + clay content with the unit g silt+clay kg$^{-1}$ soil or % (x-axis). Then, to

correctly quantify the maximum MAOC in the right unit, one should conduct a boundary analysis (or a quantile regression (Koenker 2005)) where the slope of the linear boundary curve is the maximum MAOC expressed in g MAOC per unit of silt+clay (*i,e,, the portion of the soil < 50 or 63 μm)*; remark here that it is not *total soil*, but *silt+clay,* which is the right unit for maximum MAOC (See Feng et al. 2013). Depending on the exact units used on the y-axis and x-axis, a conversion might have to be done to put the maximum MAOC in your preferred right unit (i.e., g C kg$^{-1}$ silt+clay or mg C g$^{-1}$ silt+clay or % C

of the silt+clay). It is also pertinent to have the boundary line go through the origin because if there are zero minerals (i.e. silt + clay), then there is, by definition, also zero MAOC; hence, a non-zero intercept is theoretically not possible. Here, we want to give an example of how the units matter: Georgiou et al. (2022) state that their data-driven maximum MAOC estimate of 86 g C kg$^{-1}$ mineral (= silt+clay) is a notable update to the model-predicted 45-50 g C kg$^{-1}$ soil reported by Cotrufo et al. (2019); but the value of 86 has the unit of g C kg$^{-1}$ silt+clay, whereas the estimate of 45-50 has the unit g C kg$^{-1}$ (total) soil and

are thus not comparable.

## 3 The right fractionation method

As for any soil C fractionation method, the separation of MAOC is operationally defined, so methodological shortcomings should always be considered when interpretating the data (Poeplau et al. 2018). For the testing of MAOC saturation, it is obviously pertinent to separate an as pure as possible mineral fraction that is not contaminated by, for example, particulate

organic carbon (POC). The separation of the mineral fraction is done by size and/or density separation after dispersing the soil into primary particles. Crucial here is that enough dispersive energy (through shaking or ultrasonic dispersion) is applied to break up all soil aggregates to release all sand-sized particles without applying too much dispersive energy that would break up POC and contaminate the mineral fraction. In that sense, Amelung and Zech (1999) did a very thorough assessment of the influence of ultrasonic dispersion on POM redistribution to the silt+clay fraction. They concluded that 3 kJ for 10 g soil is

optimal and that at ≥ 5 kJ there is disruption of POC and contamination of MAOC with POC. As an example, Begill et al. (2023) report that they used 100 J/ml for 10 g soil in 150 ml. Hence, they disrupted 10 g soil with 15 kJ, three times the level of energy recommended by Amelung and Zech (1999); thus, most likely disrupting POC and contaminating MAOC (Cotrufo et al. 2023). One could do a density flotation to remove contaminating POC to get a more pure MAOC, but it is our experience that one cannot float all of the POC without starting to float also some MAOC because there is a continuum of densities from

POC to MAOC. Thus, we recommend to limit the dispersive energy to the level recommended by Amelung and Zech (1999) and, most importantly, to always perform a visual inspection with a microscope of the POC fraction to assure the break-up of all soil aggregates (not concretions; this is specifically important in tropical soils that have very stable concretions (pseudo-



sand) that cannot be broken up without POC contamination of MAOC ) and of the MAOC to assure a limited (because none is impossible) contamination of the MAOC with POC (especially in high POC soils).

## 4 The right soil type

When testing the limits of mineral stabilization, confounding influences of other stabilization factors should be avoided. For example, soil types characterized by anoxic conditions such as Histosols, Gleysols, Stagnosols, Umbrisols and Cryosols should not be included in an analysis of saturation of MAOC, even if they are currently drained or melted. In such soils, MAOC has at least in part accumulated due to oxygen limitation and not through mineral stabilization. We are not stating that mineral

stabilization does not occur or is mechanistically different in such soils, but the anoxic conditions confound the effect of stabilization by interaction with the minerals. Furthermore, these soil types often have high POC levels and thus an increased chance of contamination of MAOC with POC during dispersion (Cotrufo et al. 2023). Anthrosols and Technosols are also no suitable soils to assess a "natural" limit because of all the anthropogenic processes that are/were at play. Andosols should probably be treated in their own category,due to their specific soil properties (i.e. high allophane and pyrophosphate extractable

Al) (See Beare et al. (2014) for a discussion on the soil C stabilization mechanisms in Andosols). For the same reason, also Alisols could be considered in their own category (or together with Andosols) for a saturation limit analysis. For Calcisols and Gypsisols, one obviously has to remove all inorganic carbon before determining MAOC, but this can lead to a high uncertainty in the estimate of MAOC; hence caution should be applied. Of particular importance is that samples with geogenic C are not included (see Kalks et al. 2021 for the prevalence of geogenic carbon in soils). Lastly, buried soil layers, as found in

Colluvisols, should probably not be included due to their current or past conditions that can lead to artificially high levels in MAOC. One could start thinking that the saturation concept is then limited to few soils, but the other 21 (out of 32) WRB Soil Groups do dominate across the globe and the soil types indicated to be not included in the analysis are limited in range and often associated/inclusions within specific positions within the landscape (except Cryosols but those are not often managed and thus not suitable targets to increase soil C content); even Gleysols cover only ~5% of the globe.

## 5 The right mineralogy

As has been done in almost all studies estimating a maximum of MAOC (Hassink 1997; Six et al. 2002; Feng et al. 2013; Matus 2021; Georgiou et al., 2022), the separation of soils with different mineralogies is necessary because of their differences in reactivity and thus potential to stabilize MAOC; as we indicated specifically for Andosols in the previous section. In that sense, it would be very interesting to have estimates of saturation level per clay/mineral type, because then we could define

for each soil a saturation level based on the amount of specific minerals present in the soil. However, the currently available data does not allow for that and thus a separation between the broad categories of 1:1 versus 2:1 clay dominated soils is all



that can be done. However, there are recent efforts to map clay-sized mineral distributions across the globe (See Ito & Wagai 2017) and hopefully soon the resolution of these maps will be fine enough for C saturation analyses.

## 6 The right saturation level

Since many soils are not receiving high enough C inputs to come close to soil C saturation, particularly subsoils, it is important to be cognizant of what level of MAOC can be reached for a certain silt + clay content. A search through the literature leads to three publications that estimated the maximum MAOC for 2:1 mineral dominated soils with the "right measures" and "right units" as outlined above: Feng et al. (2013) estimated $84 \pm 4$ g C kg$^{-1}$ silt+clay, Georgiou et al. (2022) estimated $86 \pm 9$ g C kg$^{-1}$ silt+clay, and Matus (2021) estimated 81 g C kg$^{-1}$ silt+clay *(note: Matus (2021) estimated it for all mineralogies and did*

*have a small intercept but samples at the boundary line were 2:1 dominated)*. Although there is overlap in data between the different studies the similarity is striking, and they average to a value of $84 \pm 3$ g C kg$^{-1}$ silt+clay. Very important to note here is that Beare et al. (2014) also estimated a maximum MAOC for allophanic (153 g C kg$^{-1}$ silt+clay) versus non-allophanic soils (116 g C kg$^{-1}$ silt+clay), but the non-allophanic soils include Gleysols and some vitric Andosols. Thus, it is not surprising that they have a higher estimate. Nevertheless, their very high estimate for the allophanic soils confirms that they should be treated

in their own category. For 1:1 mineral dominated soils, we found two very similar estimates (in part because of an overlap in datapoints between the two studies): $43 \pm 4$ g C kg$^{-1}$ silt+clay (Feng et al. 2013) and $48 \pm 6$ g C kg$^{-1}$ silt+clay (Georgiou et al. 2022); averaging to $46 \pm 4$ g C kg$^{-1}$ silt+clay. Here, we also want to give two examples of how considering the right saturation level is important. First, Salonen et al. (2023) indicate that their soils sequester more MAOC than is suggested by saturation estimates and that the relationship between total soil OC and MAOC remained linear without a flex point. A look at their data

shows that they are measuring a maximum MAOC of 60 g C kg$^{-1}$ total soil for a 2:1 mineral dominated soil with 80% silt+clay (of which actually 68% clay). If we consider the average estimate for maximum MAOC in 2:1 mineral dominated soils of 84 g C kg$^{-1}$ silt+clay and the 80% silt+clay in their soil, then we estimate saturation to be at 67 (=84*80) g C kg$^{-1}$ total soil of MAOC. Hence, it is no surprise that Salonen et al. (2023) did not see saturation of the MAOC because their soils were below the limit. Second, Schweizer et al. (2021) argued, based on the observation of a higher MAOC (expressed in the right unit of

mg C g$^{-1}$ fraction) in soils with low clay content and an observed patchy and piled up structure of SOM binding to clays, that clay content is not necessarily a limiting factor for MAOC storage. However, all their reported MAOC contents are below 84 mg C g$^{-1}$ fraction (see Fig. 1e in their publication) and thus perfectly align with our estimate of maximum MAOC.

Lastly, to show that getting the rights right is important, we used the dataset (available at https://doi.org/10.5281/zenodo.7966076) of Begill et al. (2023), who suggested that there is no maximum MAOC content.

Given that their dataset is based solely on German soils, the right mineralogy is 2:1 with a maximum MAOC of $84 \pm 3$ g C kg$^{-1}$ silt+clay. Given this, 159 out of their 189 soils are below the MAOC saturation limit; only 30 (<16% of all samples) are "above" the saturation limit. Of those 30 soils, 13 are Gleysols, 4 are Stagnosols, 3 are Anthrosols, and 1 is a "miscellaneous" soil; 1 soil is a Regosol with an extremely high silt+clay content (97%) while not being a well-developed soil; hence, it probably





contains geogenic carbon. Thus, 8 soils remain above the limit after considering the "right soil type", which is 4% of the soils

in their dataset. Furthermore, 7 out of those soils are below 91 g C kg$^{-1}$ silt+clay, which is within one standard deviation of the value of Georgiou et al. (2022), i.e. exactly what you would expect in a normal distribution of samples/errors. However, more interesting is that when all Gleysols, Stagnosols, Anthrosols and "miscellaneous" soils are taken out of the dataset and a boundary analysis is done, we find a perfect boundary with an estimated maximum MAOC of 78 ± 6 g C kg$^{-1}$ silt+clay (see Fig. 1); a very similar estimate to the others for 2:1 dominated soils. Thus, supporting the use of the rights to estimate rightly

the maximum mineral stabilization. Including this additional estimate to the above three examples leads to an average for 2:1 mineral dominated soils of 82 ± 4 g C kg$^{-1}$ silt+clay or thus roughly 8% C in the mineral fraction.

## Conclusions

In conclusion, we can confidently state that 1) there is strong empiric evidence for a maximum of C stabilization by minerals and 2) without more mineralogy data, the estimate of 82 g C kg$^{-1}$ silt+clay for 2:1 clay dominated soils is rather solid and with

some less confidence 46 g C kg$^{-1}$ silt+clay for 1:1 clay dominated soils. Furthermore, from a practical standpoint, we conclude that:

1. Most of our "managed" soils have a saturation deficit, and not a small one.
2. We should focus our efforts to sequester C (both as MAOC and POC) on soils with a high silt+clay content that are far from the maximum MAOC.

3. In sandy soils, we should focus on how to get POC stored because the MAOC will be saturated fairly quickly (unless they have anoxic conditions).
4. When estimating potentials for rates and amounts of sequestering soil C with models, saturation dynamics should be considered (see Stewart et al. 2008; Georgiou et al. 2022) and made spatially explicit based on environmental conditions.




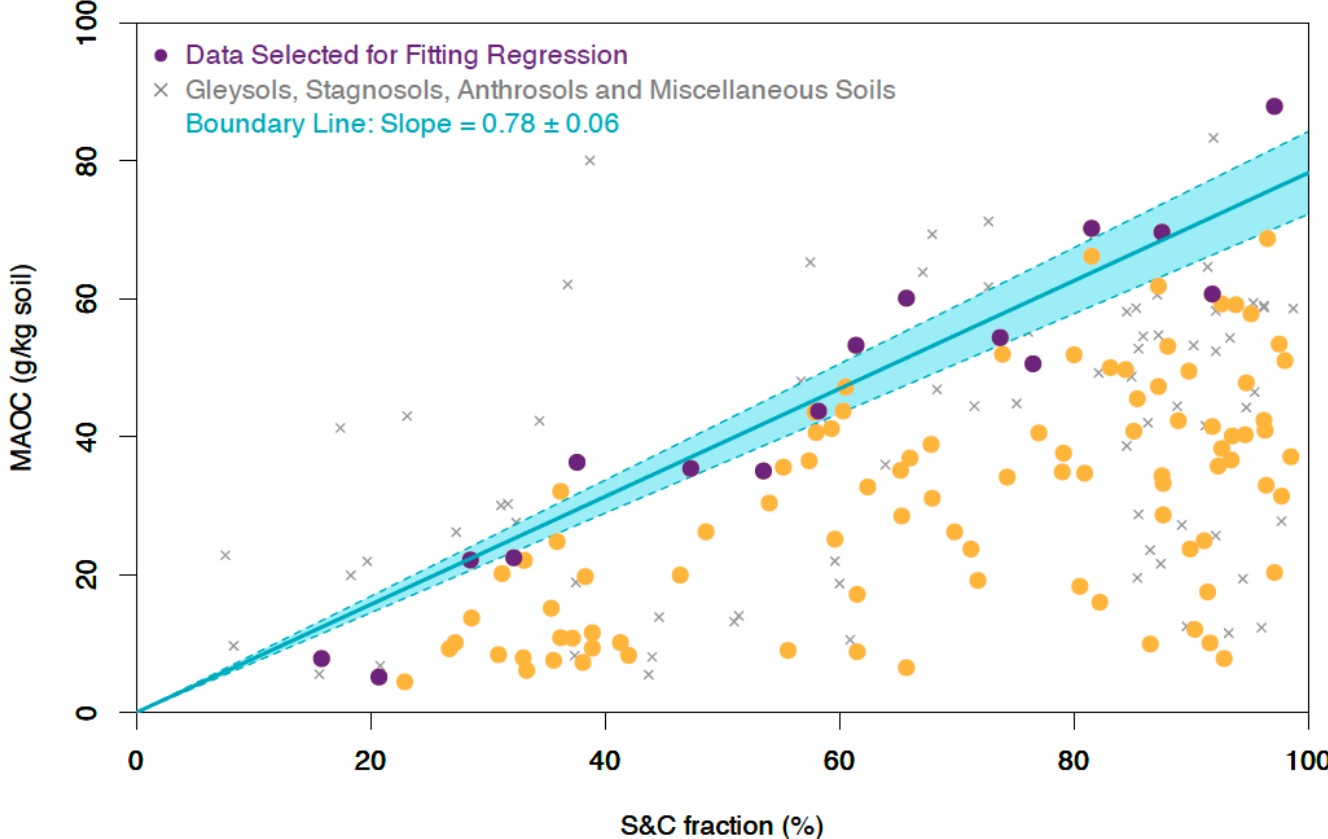

**Figure 1: Boundary analysis of mineral associated organic C (MAOC) versus silt & clay content of the fine soil based on the Begill et al. (2023) dataset but without Gleysols, Stagnosols, Anthrosols, and "miscellaneous" soils (indicated by grey crosses). Boundary line (blue line) based on the 90[th] percentile (purple dots) with confidence interval (blue shade). Orange dots are all other soils in the dataset, including Cambisols, Luvisols, Phaeozems, Regosols, and Vertisols. Note that much weaker trend between MAOC and the silt & clay content is visible for the subgroup of soils with grey crosses, indicating the presence of non-texture dependent factors (e.g. anoxic conditions) determining the measured MAOC content in those soils.**

The contact author has declared that none of the authors has any competing interests.

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
