# Peer review of "The six rights of how and when to test for soil C saturation"

_EGUsphere, 2023_

## Author Response (AR1)

**Response to reviews for egusphere-2023-2221**

**Referee #1:**

*The authors have compiled a comprehensive overview of the MAOM-C saturation theory, shedding light on associated pitfalls and contributing to the ongoing, heated scientific debate.*

*While the manuscript is generally well-written, it is not immune to certain fallacies that warrant careful consideration before publication. Consequently, I recommend the article for publication after undergoing minor revisions.*

**Response:** We thank the reviewer for this general positive assessment of our manuscript and address the concerns expressed by the reviewer below.

*Comment 1.1 l.40: The rationale behind the necessity for the boundary line to intersect at 0 is a matter of perspective. While it is undeniably accurate to assume that MAOM cannot exist without silt and clay-sized particles, the authors subsequently draw attention to impurities within the MAOM fractions, particularly micro-POM that has passed through the sieve. Consequently, the y-intercept could potentially serve as a valuable indicator of methodological errors. In any case, careful consideration is required to determine whether the linear function should be modified to favor a 0 intercept, whether linearity should be compromised in favor of a better-fitting logarithmic model, or if the linear prediction should be confined solely to the calibrated range.*

**Response:** We fully agree with the reviewer that a comparison of the boundary line without versus with an y-intercept can give an indication of how much POM has contaminated the MAOM. If there is a significant y-intercept, then it does indeed indicate that contamination occurred and those samples predominantly determining the y-intercept should be omitted from the boundary line analysis. We also agree that a linear prediction should be confined to the range of clay+silt within the samples; predicting beyond that range is prone to high uncertainty. However, we do not agree that a logarithmic model should be considered when the right measures with the right units are used. It does not make mechanically sense that the capacity to stabilize carbon by the silt+clay fraction is dependent on the amount of silt+clay in the soil. To address all of the above, we have included the following sentences: "It is also pertinent to have the curve 1) not predict beyond the range of available silt+clay measurements, 2) be linear because it does not make mechanically sense that the capacity of individual particles of the silt+clay fraction to stabilize C is dependent on the total amount of silt+clay in the soil and 3) go through the origin because if there are zero minerals (i.e. silt + clay), then there is, by definition, also zero MAOC; hence, a non-zero intercept is theoretically not possible. Nevertheless, the difference between boundary lines without and with an y-intercept could indicate the degree of contamination of MAOC with POC and thus which samples should possibly be omitted from the analysis because of too high levels of contamination."

*Comment 1.2 l.53-60: In my sincere opinion, this assertion appears to be rooted in outdated literature. Contemporary studies indicate that comparing overall energy input becomes inconclusive without accounting for additional parameters. The application of high energy over a short period does not necessarily yield comparable results to low energy applied over a longer duration, even when the total energy input remains constant. (The distinction between a pizza prepared for 1 minute at 1000°C and one cooked for 10 minutes at 100°C is significant). Notably, North et al. in 1976 demonstrated that the majority of dispersion occurs with the first impulse, which is driven by the power output of the sonotrode. This finding is corroborated by studies, such as those by Poeplau & Don 2014 (Effect of ultrasonic power on soil organic*

*carbon fractions) or Graf-Rosenfellner et al. 2018 (Replicability of aggregate disruption by sonication—an inter-laboratory test using three different soils from Germany), emphasizing the need for similar sonotrode output power to achieve comparable results. Therefore, it is not sufficient to solely compare the overall energy input in joules without acknowledging other relevant parameters. There exist more contemporary methods to standardize sonication procedures than those proposed by Amelung and Zech in 1999. It is imperative to reference a range of studies exploring the translocation of POM at various energy levels. Additionally, the efficacy of alternative dispersion methods remains questionable as they, too, are prone to errors. When delving into specific methods here, it is crucial to address the strengths and weaknesses of all commonly employed techniques.*

*It is essential to underscore that there is no universally "correct fractionation method" since these methods are always tailored to the specific soils being fractionated, accounting for the proposed concept of "the right soil type."*

**Response:** We thank the reviewer for this valuable comment and will take further care in our revision of the according section to acknowledge the additional caveats when assessing the role of energy input and the various ways of applying a known amount of energy on soil for dispersion. We fully agree also with the reviewer that there is not one correct fractionation method. And in light of this, we have changed the title of this section to "the right dispersive energy and application". We also referenced and addressed this in light of the more recent literature suggested by the reviewer. However, within the limits of this letter to SOIL we can unfortunately not delve into all the specifies, but we have strengthened our statements add to this section the following sentence: "Thus, even though the ideal dispersive energy is soil dependent, we recommend to limit the dispersive energy to the level recommended by Amelung and Zech (1999) and, most importantly, to always perform a visual inspection with a microscope of the POC fraction to assure the break-up of all soil aggregates (not concretions; this is specifically important in tropical soils that have very stable concretions (pseudo-sand) that cannot be broken up without POC contamination of MAOC ) and of the MAOC to assure a limited (because none is impossible) contamination of the MAOC with POC (especially in high POC soils). The principals of Amelung and Zech (1999) still hold up to this day, despite the fact that more recent works (e.g.Poeplau & Don 2014; Graf-Rosenfellner et al. 2018) did emphasize that even if one stays below the recommended maximum energy input, results between application of the same total amount of energy input but at varying intensities and length can still significantly affect the overall result of a soil carbon fractionation analysis."

***Comment 1.3 l.57:*** *I find it challenging to trace the citation from Cotrufo et al. 2023, as their study did not specifically investigate the aspect in question. Cotrufo et al. 2023 refers to Leuthold et al. 2022, which is a non-peer-reviewed book chapter. Leuthold et al. 2022, in turn, cites Oorts et al. 2005 and Amelung & Zech 1999; however, the energy values mentioned by Leuthold et al. 2022 cannot be located in the literature they referenced. It is essential to note that while there may be literature addressing the enrichment of POM in the MAOM fraction through sonication, it is advisable to cite original studies for accuracy and reliability.*

**Response:** We have removed the reference to Cotrufo et al. 2023 because this particular statement is actually a conclusion based on the preceeding rational for too high levels of energy used. Furthermore, in general, we fully agree that original studies should be cited instead of more recent papers that are simply based on these original studies and we have done so in the revised version of the MS.

***Comment 1.4 l.124:*** *The authors should provide an explanation for the basis of the assumption that the organic carbon in the mentioned soil is of geogenic origin.*

**Response:** In the text, we do not ascertain that the carbon is of geogenic origin, but we suggest that it might be. We acknowledge that it cannot be clear in a Regosol if high C is derived from active sequestration as soil organic C or rich due to legacy effects such as geogenic C present in the clay rich sedimentary parent rock. Hence, it cannot be excluded that the soils contains geogenic carbon and therefore should be excluded in the analysis. We have changed the text to reflect this: "1 soil is a Regosol with an extremely high silt+clay content (97%) while not being a well-developed soil; hence, it cannot be excluded that this soil contains geogenic carbon stemming from the (likely) sedimentary rock parent material."

**Referee #2**

*The authors provide an assessment of MAOC saturation, emphasizing the importance of selecting appropriate measures, units, fractionation methods, soil types, mineralogy, and saturation levels. The manuscript is well written and offers a succinct yet thorough contribution to the consolidation of the MAOC saturation theory. Overall, this article provides valuable insights for researchers engaged in soil carbon studies and ensures a nuanced approach to saturation limit analyses. I recommend to accept this paper after minor revisions.*

**Response:** We thank the reviewer for this very positive assessment of our manuscript and address the minor concerns expressed by the reviewer below.

The following is a critical review of the key points in the paper:

***Comment 2.1 The right measures.*** *This section provides well supported, logical arguments for the limitations of plotting soil C vs C input and plotting MAOC vs total SOC for estimating Soil C saturation limits. However, the claim that plotting MAOC versus silt+clay content is the "most elegant, accurate, and preferable way" to test for saturation is stated without robust evidence or comparative analysis. Adding more empirical support or a comparative discussion would bolster this assertion. Also, while the section mentions confounding factors, it does not delve into specific details about how these factors might impact the accuracy of assessing soil C saturation. Providing examples or elaborating on these confounding factors would strengthen the argument.*

**Response:** We agree with the reviewer that providing more empirical evidence that plotting MAOC vs. silt+clay content is the way to go is necessary and still needs to be provided for many soil types. What we actually meant was that the most elegant way to assess saturation for a single soil type with a single texture at a single climate would be plotting C input vs MAOC. As a working hypotheses that delivers good results in practical terms (without interpreting the underlying mechanisms of stabilization), this way of assessing the MAOC stabilization potential has provided valuable guidance and can still serve as a guideline for readers. If there is MAOC saturation, the positive relation between C inputs and MAOC content should disappear above a certain level of C inputs (no further increase in MAOC with increasing C inputs). However, since all datasets that are used to assess the saturation in practice contain a large range of soil textures, mineralogies and climates, and C input is usually not known, the most practical way to test saturation is plotting MAOC vs silt+clay content. We have altered the text accordingly to make this clearer: "Soil C saturation is by definition a non-responsiveness of soil C content upon an increase of C inputs and thus should ideally be assessed by plotting soil C (or any fraction of it, e.g., mineral-associated organic C (MAOC)) versus C input (Six et al. 2002). However, most datasets used to test saturation contain a large range of soil textures, clay types and climates, and C input from vegetation to soil (especially to specific soil C fractions) is usually not known; thus, the resulting plots of soil C versus C input are

often not sensible and thus inconclusive (see West & Six 2007; Stewart et al. 2007; Feng et al. 2014). Hence, the most practical way to test saturation is plotting MAOC vs silt+clay content as a proxy for the reactive mineral phase in soil." Unfortunately, the length limitations of a SOIL letter do not allow us a further comparative discussion as suggested by the reviewer.

*Comment 2.2 L24: Consider rephrasing to "difficulties to accurately determine C inputs"*

**Response:** We changed this sentence to: "C inputs from vegetation to soil (especially to specific soil C fractions) is usually not known."

*Comment 2.3 The right units. This section provides an important discussion of units and emphasizes the lack of comparability among estimates obtained using different units. While this might seem apparent, it is a frequently overlooked aspect that has, in some instances, led to inaccurate conclusions. The acknowledgment of this potential pitfall will hopefully provide some much-needed clarity to this debate moving forward.*

**Response:** We would like to thank the reviewer for this confirmation of the usefulness of this section.

*Comment 2.4 L40: While I agree that a non-zero intercept is theoretically impossible, it could be beneficial for the authors to briefly address the potential reasons for a non-zero intercept, such as measurement inaccuracies, procedural errors, or unexpected influences on the MAOC assessment. This additional clarification would provide a more comprehensive understanding for readers and demonstrate a nuanced consideration of potential limitations in the proposed methodology.*

**Response:** As indicated to reviewer 1, we agree that a non-zero intercept indicates an error, most likely due to contamination of POM in the MAOM fraction. We have acknowledged this in the text by pointing out how comparing the boundary analysis without and with an intercept can lead to insights into potential errors. See response to reviewer 1 and line 44-46.

*Comment 2.5 The right fractionation method. This section effectively addresses the critical importance of dispersal energy in soil C fractionation . However, there is potential for enhancement by expanding the discussion to include other relevant methodological considerations, such as grinding methods and inorganic C quantification.*

**Response:** We agree that there are other methodological considerations. However, it is evident that the dispersal energy is the biggest methodological issue to be considered. Furthermore, we do acknowledge the issue of inorganic C quantification in the "right soil type" section (see L 85-87). Due to strict space limitation for SOIL letters, we cannot address some of these other methodological issues that we also think are not as big of an issue. We hope that the reviewer is satisfied with this.

*Comment 2.6 L59: Do you have any evidence to support the claim? I imagine it is correct that there is a continuum of densities, but is it true that the contamination of heavy POM into MAOM during density fractionation is greater than the contamination of fine POM into MAOM during size fractionation? What about combined size/density to isolate a distinct, fine heavy fraction? While the accuracy of this assumption is presumed, empirical evidence supporting the claim would fortify its validity.*

**Response:** We agree (and also alluded to this in the manuscript) that a combination of size and density is useful, but one just simply has to realize that a full separation of POM and MAOM is not possible. In the past we have obtained empirical evidence for this, but this was unfortunately never published because it never fit within a manuscript. Hence, we can only state that "it is our experience" in line 64-65.

***Comment 2.7 The right soil type.*** *This section articulates the exclusion criteria for certain soil types effectively. It emphasizes the need to avoid confounding factors, such as anoxic conditions and anthropogenic influences, which could impact the interpretation of MAOC saturation. The detailed categorization of soil types and the rationale for their exclusion or specific treatment enhances the robustness of the approach and demonstrates a nuanced understanding of methodological challenges.*

**Response:** We would like to thank the reviewer for this positive evaluation of this section and finding it very useful.

***Comment 2.8 The right mineralogy.*** *In this section, a more detailed exploration of concrete applications integrating mineralogy for the estimation of saturation limits would enhance the depth of the discussion. Identifying specific knowledge gaps and delineating a roadmap for advancing research in this domain would contribute significantly. What metrics of mineralogy would be useful to measure to improve global databases (e.g. oxalate-extractable iron and aluminum, exchangeable calcium, etc.). Addressing the collective need for a standardized approach to mineralogical considerations within the field becomes crucial. A prospective avenue involves adopting a soil matrix capacity index, as proposed in the work by King et al. (2023), accessible at https://rdcu.be/duiyH. This proposal warrants exploration for its potential to unify methodologies and provide a standardized framework for incorporating mineralogical factors into the assessment of saturation limits.*

**Response:** While fully agreeing with the reviewer that a more detailed section on how to exactly consider mineralogy within C saturation studies would be very beneficial, the space/word limitations of SOIL Letters does not allow to include such a very interesting section. Nevertheless, based on the comment of the reviewer, we also realized that given the data scarcity, it is currently probably most important to consider the right clay type and therefore have adjusted the title and text to emphasize the right clay type. See line 105-113.

***Comment 2.8 The right saturation level.*** *This section elucidates the importance of employing accurate parameters and categorizations to derive meaningful estimates of maximum mineral stabilization, emphasizing the need for precision in studying MAOC saturation limits.*

*The authors effectively articulated their key points, substantiating them through persuasive back-of-the-envelope calculations. Their arguments are well-supported, aligning with existing research in the field, and effectively challenging Begill et al assertion of the absence of a saturation limit. There is no additional input needed for this section, as the authors have successfully demonstrated the validity and coherence of their claims.*

**Response:** We would like to thank the reviewer for this very positive evaluation of this section.

---

## Author Response (AR2)

**Response to Executive Editor review for egusphere-2023-2221**

The first author addressed all comments by the Executive Editor and conducted a last thorough review of the whole text. All co-authors also performed a last review to assure that the whole text has all the dots on the I's. Below is the text with track changes to indicate all last changes made.

[revised manuscript text omitted]